# Histopathological Characterization of Abdominal Aortic Aneurysms from Patients with Multiple Aneurysms Compared to Patients with a Single Abdominal Aortic Aneurysm

**DOI:** 10.3390/biomedicines11051311

**Published:** 2023-04-28

**Authors:** Daniel Körfer, Philipp Erhart, Susanne Dihlmann, Maani Hakimi, Dittmar Böckler, Andreas S. Peters

**Affiliations:** 1Department of Vascular and Endovascular Surgery, University Hospital, 69120 Heidelberg, Germany; philipp.erhart@med.uni-heidelberg.de (P.E.); susanne.dihlmann@med.uni-heidelberg.de (S.D.); dittmar.boeckler@med.uni-heidelberg.de (D.B.); andreas.peters1@med.uni-heidelberg.de (A.S.P.); 2Department of Vascular Surgery, Lucerne Cantonal Hospital, 6000 Lucerne, Switzerland; maani.hakimi@luks.ch; 3Vaskuläre Biomaterialbank Heidelberg (VBBH), University Hospital, 69120 Heidelberg, Germany

**Keywords:** abdominal aortic aneurysm, vascular diseases, molecular pathology, interleukin-1beta

## Abstract

The aim of this study was to investigate histopathological differences in abdominal aortic aneurysms (AAAs) between patients with multiple and single arterial aneurysms, as we suspect that there are different underlying mechanisms in aneurysm formation. Analysis was based on a previous retrospective study on patients with multiple arterial aneurysms (*mult-AA*; defined as at least four, *n* = 143) and a single AAA (*sing-AAA*, *n* = 972) who were admitted to our hospital for treatment between 2006 and 2016. Available paraffin-embedded AAA wall specimens were derived from the Vascular Biomaterial Bank Heidelberg (*mult-AA*, *n* = 12 vs. *sing-AAA*, *n* = 19). Sections were analyzed regarding structural damage of the fibrous connective tissue and inflammatory cell infiltration. Alterations to the collagen and elastin constitution were assessed by Masson–Goldner trichrome and Elastica van Gieson staining. Inflammatory cell infiltration, response and transformation were assessed by CD45 and IL-1β immunohistochemistry and von Kossa staining. The extent of aneurysmal wall alterations was assessed by semiquantitative gradings and was compared between the groups using Fisher’s exact test. IL-1β was significantly more present in the tunica media in *mult-AA* compared to *sing-AAA* (*p* = 0.022). The increased expression of IL-1β in *mult-AA* compared to *sing-AAA* indicates inflammatory processes play a role in aneurysm formation in patients with multiple arterial aneurysms.

## 1. Introduction

Despite the highly established diagnostic and therapeutic standards, arterial aneurysms continue to pose a life-threatening hazard for affected patients. Clinical observations have led to the assumption that patients with multiple arterial aneurysms differ in their course of disease from patients with a single abdominal aortic aneurysm (AAA), both natural and post-interventional. Few prior studies have addressed this subject. However, for patients with multiple aneurysms, more subsequent aneurysms and higher complication rates for thrombotic and embolic events have been described [1,2,3]. Following the hypothesis that etiological mechanisms of multiple arterial aneurysms might differ from those of a single abdominal aortic aneurysm, we investigated clinical characteristics of both patient groups in a recently published study. Patients with multiple aneurysms were younger at initial diagnosis compared to patients with a single abdominal aortic aneurysm and presented a distinct iliac, femoral and popliteal aneurysm distribution [4].

In AAA formation, proteolytic and chronic inflammatory processes lead to vessel wall degeneration. The infiltration of inflammatory cells along with the interaction of the vascular smooth muscle cells (VSMCs) and the extracellular matrix (ECM) reduces vessel wall stability and functionality, eventually causing aneurysm formation and, consequently, rupture [5,6]. The purpose of this study is to investigate the histopathological characteristics of aneurysm wall specimens to support the hypothesis of differences between multiple and single arterial aneurysm formations and to provide possible molecular target areas for further research.

## 2. Materials and Methods

The selection and characteristics of patients enrolled in this study have been described previously [4]. In the referred retrospective study, 2189 patients diagnosed with a true arterial aneurysm in our hospital from 2006 to 2016 were analyzed. Patients with arterial pathologies other than a true aneurysm (dissection, intramural hematoma, penetrating aortic ulcer or false aneurysm) were excluded. Patients with a diagnosis of a hereditary disorder of the connective tissue (e.g., Marfan syndrome, Ehlers–Danlos syndrome or Loeys–Dietz syndrome) or vasculitis were also excluded. Multiple arterial aneurysms (at least four) were found in 143 patients (the *mult-AA* group), whereas 972 patients had a single abdominal aortic aneurysm (the *sing-AAA* group).

Paraffin-embedded abdominal aortic wall specimens of patients from both groups were derived from the Vascular Biomaterial Bank Heidelberg (VBBH) [7]. Respective sections were analyzed regarding structural damage of the fibrous connective tissue and inflammatory cell infiltration. Processing, staining (Masson–Goldner trichrome, Elastica van Gieson and von Kossa) and immunohistochemical staining (CD45 and IL-1β) were performed using standard procedures. The following primary antibodies were used: mouse-anti-human CD45 (1:1000; #3575; pan-leukocyte marker; Cell Signaling Technology, Inc., Danvers, MA, USA) and mouse-anti-human IL-1β (1:300; #12242; IL-1β protein marker; Cell Signaling Technology, Inc.). Staining and immunohistochemistry protocols are available upon request. Briefly, 4 μm sections were deparaffinized, rehydrated and incubated in 100 mM citrate buffer with a pH of 6.0 for antigen retrieval prior to incubation with the primary antibody at 4 °C overnight. After washing, detection was performed by using the Dako REAL Detection System, peroxidase/AEC rabbit/mouse (DAKO; Agilent Technologies, Inc., Santa Clara, CA, USA), according to the recommendations of the manufacturer. All sections were counterstained with hematoxylin. The extent of aneurysmal wall alterations was assessed by semiquantitative grading (Table 1). Established grading classifications were applied and partially extended [8,9]. Analysis was performed by a single observer under supervision. The extent of wall alterations was dichotomized and compared between the two groups using Fisher’s exact test.

Atherosclerotic risk factors, comorbidities, age at time of surgery, maximum AAA diameter, and preoperative C-reactive protein (CRP) and leukocyte blood levels of patients with available aortic wall specimens were assessed and compared between the two groups using the Mann–Whitney U test and Fisher’s exact test.

*p*-values < 0.05 were considered statistically significant. Analysis was conducted using GraphPad Prism 9.3.1. Statistical consultation was provided by the Institute of Medical Biometry at the University of Heidelberg. The study protocol was approved by the Ethics Committee of the Medical Faculty of the University of Heidelberg (file reference: S-452/2016 and S-301/2013).

## 3. Results

From our previous study, AAA wall specimens were available from 12 patients of *mult-AA* and 19 patients of *sing-AAA* [4]. All patients were male. There was no difference regarding atherosclerotic risk factors, comorbidities, age at time of surgery, maximum AAA diameter, and preoperative CRP and leukocyte blood levels between the patients in the two study groups (Table 2).

Figure 1 and Table 3 illustrate the extent of the aneurysmal wall alterations in both groups for the respective staining/immunohistochemistry.

According to the Masson–Goldner trichrome staining, there was no difference regarding collagen degradation (*p* = 0.705, Figure 2a). An absence of elastic fibers was observed twice more frequently in samples from the *sing-AAA* group than in the *mult-AA* group (*p* = 0.130, Figure 2b). The extent of calcification in the tunica media did not differ between the groups (*p* = 1, Figure 2c). CD45 immunostaining revealed that *sing-AAA* samples contained twice the amount of high-grade lymphocyte infiltrations than *mult-AA* samples (*p* = 0.262, Figure 2d). IL-1β was significantly more present in the tunica media of AAAs in patients in the *mult-AA* group compared to patients in the *sing-AAA* group (*p* = 0.022, Figure 2e).

## 4. Discussion

In this study, histopathological differences in abdominal aortic aneurysm wall specimens between patients with multiple (at least four) arterial aneurysms and patients with a single AAA were investigated. Our data indicate that IL-1β expression in the AAA wall is increased in patients with multiple arterial aneurysms compared to that in patients with a single abdominal aortic aneurysm. Moreover, more degradation of elastic fibers was present in the single AAA group, but was statistically not significant. The two patient groups did not differ regarding atherosclerotic risk factors, comorbidities, age at surgery, AAA size, or CRP and leukocyte blood levels.

To our knowledge, studies of multiple arterial aneurysms are rare, with no histopathologic analyses reported thus far. The available literature includes terms such as diffuse aneurysmal disease, aneurysmosis, and arteriomegaly [1,2,10,11]. However, there is no precise definition or distinction. Likewise, the etiologic specifics of this condition have not been adequately investigated, with genetic and inflammatory influences being discussed [11,12].

The formation of abdominal aortic aneurysms is multifactorial, with inflammatory processes and the degeneration of the aortic wall resulting from the loss of structural proteins. As a result of a systemic process with additional local biomechanical stress, inflammatory processes with increased matrix metalloprotease (MMP) activity and apoptosis of vascular smooth muscle cells are observed in the vessel wall. Transmural inflammatory cell infiltration includes neutrophils, B-cells, T-cells, mast cells, macrophages and natural killer cells [13]. Elastases and MMPs degrade elastic fibers and type I and III collagen. This is enhanced by the reduced inhibition of MMPs in AAAs, which is physiologically provided by endothelial aortic cells and vascular smooth muscle cells [14]. In addition, reactive oxygen species and atherosclerosis play an important role in AAA development, whereby these and the previously mentioned etiological mechanisms interact and favor each other [15].

IL-1β is a proinflammatory cytokine. In addition to its role in the physiological immune response, it also contributes to the development of inflammatory pathologies. IL-1β induces local inflammatory processes and activates neutrophilic granulocytes, monocytes and macrophages [16]. As a result of inflammasome activity, IL-1β was shown to be involved in the innate immune response [17]. The expression of IL-1β was found to be increased in the tunica media of AAA wall specimens compared to specimens from organ donors without AAAs [18]. In addition to other inflammasome components (ASC, CASP1, NLRP3 and IFI16), IL1B gene expression is increased in AAA wall specimens compared to controls [8,19,20]. The ultimate consequence of these inflammatory reactions is the death of VSMCs and loss of elastic fibers, which leads to AAA progression and can eventually result in rupture.

Higher IL-1β levels in the tunica media of AAA specimens of patients with multiple arterial aneurysms might point to an increase in inflammasome activity compared to patients with a single AAA. However, given the complex sequence of inflammatory processes in AAAs, a corresponding interpretation can only be made with great caution. Further signaling processes involved in the development of multiple arterial aneurysms would need to be investigated.

In contrast to an increased inflammatory state in AAA specimens in the *mult-AA* group, less degeneration of elastic fibers was observed in the present study, which in turn, supports the chronically degenerative character found in the *sing-AAA* group.

A limitation of this study is the retrospective study design for the identification of patients in each group. In addition, the number of samples analyzed is very small, owing to the rare occurrence of multiple arterial aneurysms, and thus, the availability of corresponding tissue from open aneurysm surgeries. Finally, at the time of histopathological analysis—when surgical repair is indicated—the AAA is already in a far-progressed stage with correspondingly pronounced structural vessel wall alterations. Therefore, statements on the mechanism of origin can only be made with reservations.

Although specific etiologic mechanisms are suspected in the development of multiple arterial aneurysms, this does not impact the clinical management of affected patients at this current time. Nevertheless, further studies should be conducted on both etiologic differentiation and natural and postinterventional outcomes. Eventually, further insights could optimize diagnostic and therapeutic actions and timing for the affected patients.

## 5. Conclusions

The increased expression of IL-1β and less degeneration of elastic fibers in AAA wall specimens in patients with multiple arterial aneurysms compared to patients with a single AAA suggests that inflammatory rather than chronically degenerative processes play a role in aneurysm formation in patients with multiple arterial aneurysms. Different underlying pathological mechanisms could influence future therapeutic approaches. Further research is required to substantiate the results presented herein.

## Figures and Tables

**Figure 1 biomedicines-11-01311-f001:**
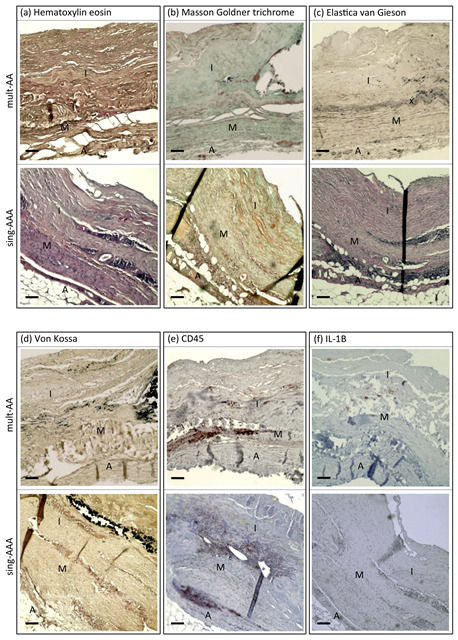
AAA wall sections of patient from mult-AA and sing-AAA groups following staining/immunohistochemistry: (**a**) hematoxylin eosin, (**b**) Masson–Goldner trichrome, (**c**) Elastica van Gieson, (**d**) von Kossa, (**e**) CD45 and (**f**) IL-1β. I = tunica intima. M = tunica media. A = tunica adventitia. x = internal elastic membrane. Scale bar, 250 µm.

**Figure 2 biomedicines-11-01311-f002:**
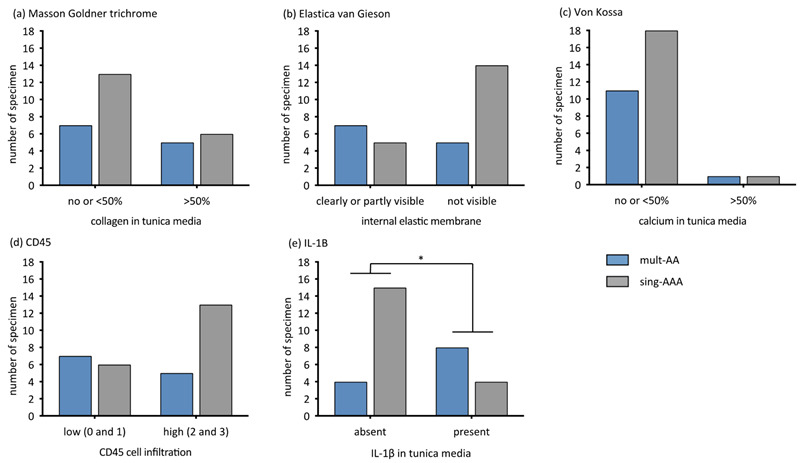
Extent of aneurysmal wall alterations compared dichotomously between mult-AA and sing-AAA groups in respective staining/immunohistochemistry by Fisher’s exact test. The *y*-axis describes the number of specimens. * statistical significance. (**a**) Extent of collagen alteration (grade 0–1 vs. grade 2). Masson–Goldner trichrome, *p* = 0.705. (**b**) Extent of internal elastic membrane alteration (grade 0–1 vs. grade 2). Elastica van Gieson, *p* = 0.130. (**c**) Extent of calcification in tunica media (grade 0–1 vs. grade 2). von Kossa, *p* = 1. (**d**) Extent of CD45-associated cell infiltration (grade 0–1 vs. 2–3), CD45. *p* = 0.262. (**e**) Extent of IL-1β expression in tunica media (grade 0 vs. grade 1). IL-1β, *p* = 0.022.

**Table 1 biomedicines-11-01311-t001:** Grading of aneurysmal wall alteration with different staining methods.

	Grade 0	Grade 1	Grade 2	Grade 3
Masson–Goldner trichrome	no collagen in tunica media	collagen in <50% of tunica media	collagen in >50% of tunica media	-
Elastica van Gieson	internal elastic membrane clearly visible	internal elastic membrane partly visible	internal elastic membrane not visible	-
von Kossa	no calcium in tunica media	calcium in <50% of tunica media	calcium in >50% of tunica media	-
CD45	no lymphocytes	lymphocytes in tunica intima and plaque	diffuse lymphocytic infiltration in tunica media and tunica adventitia	tertiary lymphoid structure in tunica adventitia
IL-1 β	absent in tunica media	present in tunica media	-	-

Definition of semiquantitative grading of different aneurysmal wall alterations for Masson–Goldner trichrome, Elastica van Gieson, von Kossa, CD45 and IL-1B staining. Grading adapted from Dihlmann et al. and Rijbroek et al. [8,9].

**Table 2 biomedicines-11-01311-t002:** Clinical characteristics of patients in mult-AA and sing-AAA groups.

	mult-AA	sing-AAA	Relative Risk (95% CI)	p
Hypertension	12 (100%)	19 (100%)	1 (0.76–1.20)	1
Diabetes mellitus	2 (16.7%)	2 (10.5%)	1.58 (0.30–8.06)	0.630
Dyslipidemia	9 (75%)	15 (78.9%)	0.95 (0.57–1.41)	1
Smoking	8 (44.4%)	10 (52.6%)	1.27 (0.67–2.30)	0.484
Coronary artery disease	7 (58.3%)	11 (57.9%)	1.01 (0.51–1.83)	1
Peripheral artery disease	3 (25%)	4 (21.1%)	1.19 (0.33–4.00)	1
Mean age at surgery (y) ± SD	74.25 ± 6.34	72.47 ± 7.03	-	0.582
Mean AAA diameter (mm) ± SD	54.17 ± 13.46	58.74 ± 8.75	-	0.247
Preoperative CRP level (mg/L) ± SD	12 ± 28.14	8.86 ± 18.46	-	0.513
Preoperative leukocyte count (/nl) ± SD	7.43 ± 1.44	8.63 ± 3.71	-	0.490

Atherosclerotic risk factors and comorbidities, mean age at surgery in years, mean maximum diameter of the abdominal aortic aneurysm (AAA) in millimeters, preoperative C-reactive protein (CRP) level in milligram per liter and preoperative leukocyte count per nanoliter of patients in mult-AA and sing-AAA groups. Atherosclerotic risk factors and comorbidities compared by Fisher’s exact test; age at surgery and maximum aneurysm diameter compared by Mann–Whitney U test.

**Table 3 biomedicines-11-01311-t003:** Extent of aneurysmal wall alterations in mult-AA and sing-AAA groups with different staining methods.

	Group	Grade 0	Grade 1	Grade 2	Grade 3
Masson–Goldner trichrome	mult-AA	1	6	5	-
	sing-AAA	0	13	6	-
Elastica van Gieson	mult-AA	4	3	5	-
	sing-AAA	0	5	14	-
Von Kossa	mult-AA	1	10	1	
	sing-AAA	4	14	1	
CD45	mult-AA	0	7	4	1
	sing-AAA	1	5	12	1
IL-1β	mult-AA	4	8	-	-
	sing-AAA	15	4	-	-

Number of aneurysmal wall specimens for each grading classification in mult-AA and sing-AAA groups for each staining/immunohistochemistry applied.

## Data Availability

Data are available in the manuscript and upon personal request to the corresponding author.

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
