# Peer review of "Histopathological Characterization of Abdominal Aortic Aneurysms from Patients with Multiple Aneurysms Compared to Patients with a Single Abdominal Aortic Aneurysm"

_biomedicines, 2023, doi:10.3390/biomedicines11051311_

Round 1

Reviewer 1 Report

This is a well written manuscript.  In my opinion the paper can be improved The authors should make an effort to address these or add statement in the strength and weaknesses of study if they cant

How many pathologists reviewed the slides was there operator agreement.  the number of pathologists who reviewed the slides should be stated and if more than one was there agreement in scoring 

do you have any other laboratory data available to further support that inflammation is involved in multi aneurysms such as neutrophil/ lymphocyte ratio, plt /lympocyte ratio, crp ferritin ect if available these should be compared between groups.

For the remaining patients identified in this study that did not have pathological specimens available please display pt characteristics like table 2 are the the same as those who had specimens available?

Author Response

Point 1: How many pathologists reviewed the slides was there operator agreement.  the number of pathologists who reviewed the slides should be stated and if more than one was there agreement in scoring

Response 1: Thank you for your comment. The slides were reviewed by a single observer under supervision. This has been added to the manuscript. There were no ambiguous findings observed at grading.

Point 2: do you have any other laboratory data available to further support that inflammation is involved in multi aneurysms such as neutrophil/ lymphocyte ratio, plt /lympocyte ratio, crp ferritin ect if available these should be compared between groups.

Response 2: Thank you for the interesting aspect. Regarding preoperative laboratory tests of the patients with available specimen, only CRP and leukocytes are implemented in our routine tests and therefore available for this study. These parameters were additionally analyzed and added to the study (in Table 2 and respective sections in the manuscript). No differences between the groups were observed. Possible differences in the inflammatory response between multiple aneurysms and single AAA seem not to influence systemic inflammation detectable in (our) standard blood samples.

Point 3: For the remaining patients identified in this study that did not have pathological specimens available please display pt characteristics like table 2 are the the same as those who had specimens available?

Response 3: Thank you for this comment. Comorbidities and risk factors were not reviewed for patients without available specimen in the respective study group. This would indeed be an interesting aspect to investigate. However, the rationale for the data of Table 2 was not to show differences between the two patient groups in general, but to investigate comparability of AAA specimen of both groups.

Reviewer 2 Report

This study investigated histopathological differences of abdominal aortic  aneurysms (AAA) between patients with multiple and single arterial aneurysms, suspecting different underlying mechanisms in aneurysm formation. This is a very important and interesting topic. I only have some minor suggestions for authors to further improve this manuscript. 

1 the scale bar is missing in this figure 1

2 there are some misinformation on label in figure 2, which should be corrected.

Author Response

Point 1: the scale bar is missing in this figure 1

Response 1: Thank you for this comment. The scale bar was added.

Point 2: there are some misinformation on label in figure 2, which should be corrected.

Response 2: Thank you for this advice. We clarified the labeling for (a), (b) and (c) concerning the inclusion of Grade 0.

Reviewer 3 Report

This is an interesting study that aimed to investigate histopathological differences of abdominal aortic aneurysms between patients with multiple and single arterial aneurysms, suspecting different underlying mechanisms in aneurysm formation. It was a retrospective study from a single site.

The abstract and manuscript are well written, and information appears complete. The background information introduces enough studies referenced to support the paper.

The methods are described in good detail. The sample is described well. One of the major limitations of the study, noted by the authors, is the retrospective study design and also the small number of samples analyzed. However, I consider that the results are important for clinical practice and that the article may be accepted for publication in its current form.

The English language is fine.

Author Response

Response: Thank you very much for your review.

Reviewer 4 Report

It is a concise and well-written article which goes straight to the point.

The discussion would benefit from your views in relation to clinical implications, management of the two groups of patients and future perspectives based on your findings. Would you consider early intervention in both groups or consider surveillance until size criteria are reached? Do you think the two groups should be managed differently and how? Do you think that testing should be considered routinely? These are all key and debatable issues to further develop in the discussion.

Here are some comments and suggestions.

Line 54-58: it would be more appropriate to say “Patients with other arterial pathologies than a true aneurysm (dissection, intramural hematoma, penetrating aortic ulcer or false aneurysm) were excluded previously in view of their diagnosis of connective tissue disorder (e.g. Marfan syndrome, Ehlers-Danlos syndrome, Loeys-Dietz syndrome) or vasculitis.”

Line 58-60: it would be more appropriate to say “Multiple arterial aneurysms (at least four) were found in 143 patients (mult-AA group) whereas 972 patients had a single abdominal aortic aneurysm (sing-AAA group).”

Line 61: it would be more appropriate to say “Paraffin-embedded abdominal aortic wall specimens of...”

Line 77-78: it would be more appropriate to say “...and compared between the two groups using Fisher’s exact test.”

Line 84-85: it would be more appropriate to say “...were assessed and compared between the two groups using Mann-Whitney U test and Fisher’s exact test.”

Line 115-116: it is more appropriate to say “Absence of elastic fibres was observed twice more frequently...”

Line 118-119: it is more appropriate to say “...samples contained twice more...”

Line 166-167: remove “in” and say “...compared to patients with...”

Line 178: remove “an” and say “AAA is already...”

Appropriate use of the English language.

Some minor issues detected for which suggestions given.

Author Response

Point 1: The discussion would benefit from your views in relation to clinical implications, management of the two groups of patients and future perspectives based on your findings. Would you consider early intervention in both groups or consider surveillance until size criteria are reached? Do you think the two groups should be managed differently and how? Do you think that testing should be considered routinely? These are all key and debatable issues to further develop in the discussion.

Response 1: Thank you very much for this comment. In our opinion, the results of our study do not yet allow implementation in clinical algorithms regarding diagnostics and therapy of respective patients. Different etiological mechanisms of multiple arterial aneurysms compared to single AAA could indeed affect the natural course of the disease and therefore influence the optimal timing of intervention. For this purpose, prospective studies of the natural and postinterventional course of both groups should be performed. Besides, further studies should investigate suspected etiological differences, for example genetic testing of respective patients. Assuming an inflammatory reaction to play a distinct role in multiple aneurysm formation, anti-inflammatory intervention could be a target for specific therapy. However, at this stage, these approaches do not play a role in the practical management of patients with multiple aneurysms today. This view was added to the discussion.

Point 2: Line 54-58: it would be more appropriate to say “Patients with other arterial pathologies than a true aneurysm (dissection, intramural hematoma, penetrating aortic ulcer or false aneurysm) were excluded previously in view of their diagnosis of connective tissue disorder (e.g. Marfan syndrome, Ehlers-Danlos syndrome, Loeys-Dietz syndrome) or vasculitis.”

Response 2: Thank you for your suggestion. However, this phrasing could possibly be misleading because patients with other arterial pathologies were excluded, as well as - irrespective of their arterial pathology - patients with hereditary connective tissue diseases and vasculitides in general. To clarify, we separated these two exclusion criteria into two sentences.

Point 3: Line 58-60: it would be more appropriate to say “Multiple arterial aneurysms (at least four) were found in 143 patients (mult-AA group) whereas 972 patients had a single abdominal aortic aneurysm (sing-AAA group).”

Response 3: Thank you for this advice. The corresponding phrase has been changed accordingly.

Point 4: Line 61: it would be more appropriate to say “Paraffin-embedded abdominal aortic wall specimens of...”

Response 4: Thank you for this advice. The corresponding phrase has been changed accordingly.

Point 5: Line 77-78: it would be more appropriate to say “...and compared between the two groups using Fisher’s exact test.”

Response 5: Thank you for this advice. The corresponding phrase has been changed accordingly.

Point 6: Line 84-85: it would be more appropriate to say “...were assessed and compared between the two groups using Mann-Whitney U test and Fisher’s exact test.”

Response 6: Thank you for this advice. The corresponding phrase has been changed accordingly.

Point 7: Line 115-116: it is more appropriate to say “Absence of elastic fibres was observed twice more frequently...”

Response 7: Thank you for this advice. The corresponding phrase has been changed accordingly.

Point 8: Line 118-119: it is more appropriate to say “...samples contained twice more...”

Response 8: Thank you for this advice. The corresponding phrase has been changed accordingly.

Point 9: Line 166-167: remove “in” and say “...compared to patients with...”

Response 9: Thank you for this advice. The corresponding phrase has been changed accordingly.

Point 10: Line 178: remove “an” and say “AAA is already...”

Response 10: Thank you for this advice. The corresponding phrase has been changed accordingly.